# Silver(I) and Gold(I) Monothiocarbonate Complexes: Synthesis, Structure, Luminescence

**Welni Duminy, Michael N. Pillay and Werner E. van Zyl ***

School of Chemistry and Physics, Westville Campus, University of KwaZulu-Natal, Durban 4000, South Africa; welniduminy@gmail.com (W.D.); niven.sa@gmail.com (M.N.P.)
* Correspondence: vanzylw@ukzn.ac.za; Tel.: +27-(0)31-260-3188

**Abstract:** The monothiocarbonate ligand, $[S(O)COR]^-$, is unusual and rare regarding its use in the formation of coordination compounds. Here, we report the synthesis and structures of the silver(I) and gold(I) monothiocarbonate complexes, $[\{Ag_4(SC(O)O^iPr)_2(2,2'\text{-bpy})_4\}(PF_6)_2]_n$ (**1**) and $[Au_2\{S(O)CO^iPr\}_2(dppe)]_n$ (**2**), respectively. Both complexes are coordination polymers, with **1** being cationic and **2** neutral. The uniqueness of the ligand is that it is monoanionic and contains both a 'hard' O-donor ligand and a 'soft' S-donor ligand in a O-C-S manifold with, in principle, electron delocalization across the three atoms. However, for both complexes **1** and **2**, it was found that the binding occurred exclusively through the S-donor atom, while the C=O portion remained dangling and was not involved in bonding. This bonding mode departs significantly from the symmetrical S-C-S type ligand such as dithiocarbamates. The structures were analysed and confirmed by NMR and X-ray crystallography.

**Keywords:** silver(I); gold(I); monothiocarbonate; coordination polymer; synthesis; structure





## 1. Introduction

The chemistry of late-transition metals with 1,1-dithiolato-type ligands is well established. This class of ligands include the dithiophosphonates [1,2], xanthates [3] and thiophene/dithiolenes [4], although the most predominant among them are arguably the dithiocarbamates [5–8] and dithiophosphates [9,10]. The dithiocarbamates have additionally found specific application as precursors in forming metal sulfide nanoparticles [11], whilst the dithiophosphates are commonly used to stabilize structurally determined atomically precise nanoclusters of high nuclearity, especially among the coinage metals (Cu, Ag, Au) [12–15].

Dithiocarbamates $[S_2CNR_2]^-$ are typically prepared from the reaction between a primary amine and $CS_2$ whilst dithiophosphates $[S_2P(OR)_2]^-$ are prepared from the reaction between $P_4S_{10}$ and a primary or secondary alcohol. Both these ligands are symmetrical—i.e., typically they have a mirror plane bisecting the central atom(s)—and as a result do not readily form complex isomers, but on the other hand, due to their symmetry, they aid tremendously in the single-crystal growth process, presumably due to a more efficient crystal packing that can be achieved, minimizing crystal forces. As a result, literature reports on structurally determined metal clusters with symmetrical dithiocarbamates or dithiophosphate ligands vastly exceed reports on those with unsymmetrical dithiophosphonate $[S_2PR(OR')]^-$ counterparts.

In continuing our work on 1,1-dithiolato-based ligands with the Group 11 metals [16–18] and exploring their rich photochemistry [19], we considered an interesting alternative to the 1,1-dithio-type ligands mentioned above by replacing the donor sulfur atom with a different chalcogen, i.e., oxygen (hetero O/Se or S/Se combinations have never been attempted). In this study, the coordination chemistry of the monothiocarbonate $[S(O)COR]^-$ ligand was thus developed and explored. This ligand class is closely related to that of the xanthates $[S_2COR]^-$, which also contains the familiar S-C-S bridging moiety

found in dithiocarbamates. The monothiocarbonates are arguably more challenging to prepare, requiring the reaction between an alcohol and O=C=S gas. The latter typically need to be prepared in-situ; we propose this inconvenience is one reason why this ligand is rarely used in metal complex formation. The subtle difference between O-C-S vs. S-C-S looks innocent, but preliminary reports suggest that this change can have a tremendous impact on the outcome of the formed product. The first obvious difference is based on Pearson's hard and soft acids and bases (HSAB) principle; the O donor atom is considered hard, while the S-donor is considered soft, and this will play a role in selectivity with borderline hard/soft metal centers with which it coordinates. We recently reported a system where the effect was greatly amplified; using the monothiocarbonate ligand, the structure of a solvated cluster $[Cu\{SC(O)O^iPr_2\}]_{16}\cdot2THF$ was described that can both desolvate and self-assemble in solution to form a giant metallaring, $[Cu\{SC(O)O^iPr\}]_{96}$ which is the largest metal cluster with a monothiocarbonate-type ligand reported to date [20]. A similar reaction with traditional dithiocarbamate or xanthate will not cyclicize in this manner but will remain a low nuclearity cluster. This reaction in part succeeds because the O-C bond is shorter than the C-S bond (assuming complete electron delocalisation).

In this study we focused on silver(I) and gold(I) centers and their reaction with monothiocarbonates. Prior to this study, no silver(I) complex and only two gold(I) complexes containing the monothiocarbonate ligand have been structurally characterized and reported [21].

## 2. Results and Discussion

### 2.1. The Silver(I) Complex $[\{Ag_4(SC(O)O^iPr)_2(2,2'\text{-}bpy)_4\}(PF_6)_2]_n$ *1*

To date, all attempts to synthesise and crystallise a silver(I) monothiocarbonate complex without an auxiliary ligand were unsuccessful, including the present study. In 1973, Murphy and Winter claimed to have isolated colourless crystals of the silver(I) *O*-ethylmonothiocarbonate complex, but without X-ray crystallographic verification, little can be said about its molecular structure [22]. The reaction of $[Ag(CH_3CN)_4]PF_6$ with $K[SC(O)O^iPr]$ in acetone and subsequent extraction in chloroform yielded a dark brown precipitate that could not be crystallized, and further attempts were abandoned. However, using nitrogen-donor auxiliary ligands aided in the formation of crystalline solids. Hence, one molar equivalent of 2,2′-bipyridine (bpy) was added, resulting in the formation of a solid-state luminescent white crystalline solid, identified as the coordination polymer $[\{Ag_4(SC(O)O^iPr)_2(2,2'\text{-}bpy)_4\}(PF_6)_2]_n$ *1*.

Single crystals of *1* were obtained by slow diffusion of diethyl ether vapour into a concentrated dichloromethane solution at 4 °C. Although there are a number of luminescent silver(I) complexes with S-donor ligands and short Ag···Ag interactions reported [23–31], complex *1* marks the first silver(I) monothiocarbonate complex to be structurally characterised by SC-XRD. Complex *1* crystallised as colourless long needles in the orthorhombic *Pca*$2_1$ space group. The asymmetric unit consisted of a $[\{Ag_4(SC(O)O^iPr)_2(2,2'\text{-}bpy)_4\}]^{2+}$ unit and two $PF_6^-$ anions. The molecular structure of the asymmetric unit is shown in Figure 1, and the relevant crystallographic data is summarised in Table 1.

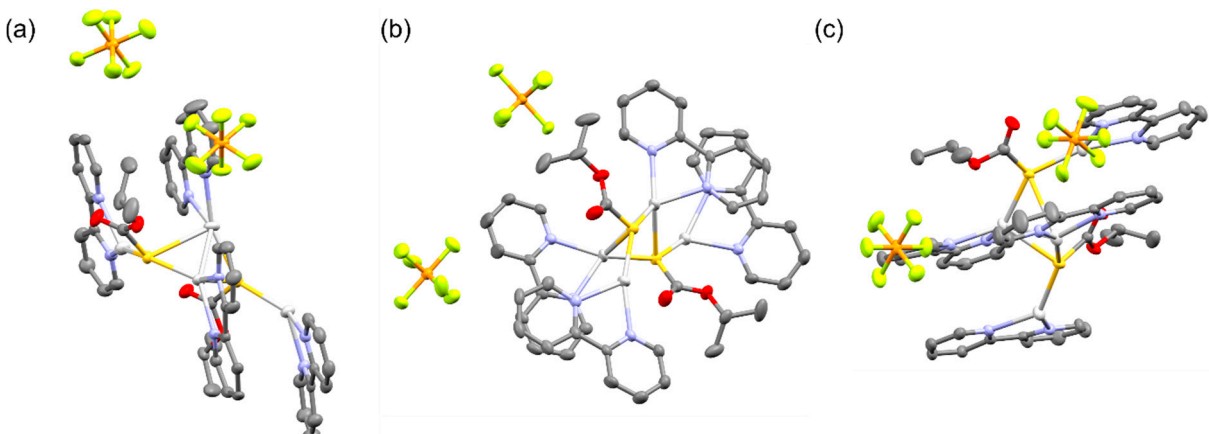

**Figure 1.** Molecular structure of **1** viewed along (**a**) the *a*-axis, (**b**) the *b*-axis, and (**c**), the *c*-axis. Thermal ellipsoids are shown at 50% probability. Hydrogen atoms have been omitted for clarity. Colour code: white = silver, yellow = sulphur, red = oxygen, pale blue = nitrogen.

**Table 1.** Crystallographic data for complexes **1** and **2**.

| Complex | 1 | 2 |
|---|---|---|
| CCDC number | 2,126,277 | 2,126,328 |
| Chemical formula | $C_{48}H_{46}Ag_4N_8O_4S_2\ 2(F_6P)$ | $C_{34}H_{38}Au_2O_4P_2S_2$ |
| $M_r$ | 1584.47 | 1030.63 |
| Crystal system | Orthorhombic | Monoclinic |
| Space group | $Pca2_1$ | $P2_1/n$ |
| Temperature (K) | 150 | 150 |
| $a$ (Å) | 28.2245 (7) | 11.7462 (9), |
| $b$ (Å) | 7.1755 (2) | 39.197 (3), |
| $c$ (Å) | 27.6678 (7) | 15.6529 (15) |
| $\alpha$ (°) | 90 | 90 |
| $\beta$ (°) | 90 | 107.882 (4) |
| $\gamma$ (°) | 90 | 90 |
| V (Å$^3$) | 5603.4 (3) | 6858.7 (10) |
| Z | 4 | 8 |
| $P_{calcd}$ (g cm$^{-1}$) | 1.878 | 1.996 |
| $\mu$ (mm$^{-1}$) | 1.600 | 8.798 |
| $T_{min}, T_{max}$ | 0.575, 0.746 | 0.385, 0.746 |
| Reflections collected | 73,431 | 56,474 |
| Independent reflections | 13,261 | 15,085 |
| Observed reflections [$I > 2\sigma(I)$] | 12,168 | 11,582 |
| $R_{int}$ | 0.029 | 0.037 |
| $R[F^2 > 2\sigma(F^2)]$ | 0.025 | 0.085 |
| $wR(F^2)$ | 0.052 | 0.215 |
| $S$ | 1.052 | 1.14 |
| $\Delta\rho_{max}, \Delta\rho_{min}$ (e Å$^{-3}$) | 1.00, −0.54 | 5.08, −3.88 |

Complex **1** has a polymeric one-dimensional chain structure that consists of (bpy)Ag···Ag(bpy) units bridged by the monothiocarbonate ligand and binding through the S-donor atom. The asymmetric unit of the crystal structure can be considered the simplest repeating unit of the polymer. The polymeric chain was cationic, and the 2+ charge of each repeating unit was balanced by two $PF_6^-$ anions, which were derived from the $[Ag(CH_3CN)_4]PF_6$ starting material. The 2,2′-bpy ligands were not directly involved in forming the polymer; these ligands stabilised the Ag(I)/S chain laterally by completing the tetrahedral coordination sphere of each Ag(I) centre (Figure 2). The monothiocarbonate ligands, on the other hand, joined adjacent Ag(I)···Ag(I) units to form an infinite $[Ag_2–S–Ag_2–S]_n$ chain. Although many polymeric Ag(I) have been reported, this is the first example of a $[Ag_2–S–Ag_2–S]_n$ type polymer.

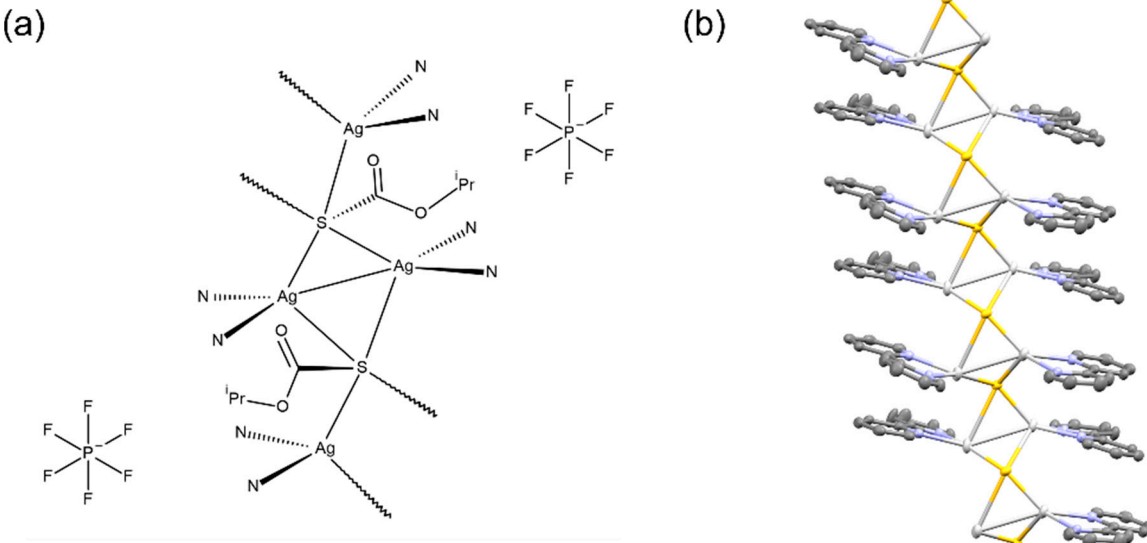

**Figure 2.** (**a**) Structure of the repeating unit in **1** (only the N atoms of the 2,2′-bpy ligands are shown) (**b**) Representation of the coordination polymer. Only the sulphur atoms (yellow) of the monothiocarbonate ligands are shown. All H atoms omitted for clarity.

Bonding of 2,2′-bpy ligands to Ag(I) centres was asymmetrical; the distances between Ag(I) and the N atoms in a given 2,2′-bpy ligand differed, and ranged from 2.272 Å to 2.356 Å, while the average Ag–N distance was 2.31 Å. The monothiocarbonate ligands adopted a tetracoordinate tetraconnective $\kappa_4$: $\mu_4$-S bonding mode (Figure 3), which has not been observed in any monothiocarbonate complexes (nor to our knowledge, in dithiocarbamate complexes). Ag–S distances were in the range 2.401–3.031 Å similar to what was reported by Zhang et al. in a cluster-based two-dimensional polymer [32]. In complex **1**, each monothiocarbonate ligand bonded to four Ag(I) centres to form a distorted square pyramid, with two short Ag(I)···Ag(I) interactions. The Ag(I)···Ag(I) distances (3.019 Å and 3.112 Å) were shorter than the sum of two van der Waals radii (3.44 Å), suggesting argentophilic interactions. It has been argued by Schmidbaur and Schier [33] that the presence of *bona fide* argentophilic Ag···Ag interactions should be done with caution. In particular, Moreno-Alcántar and co-workers studied a number of silver(I) thiolate ligands with phosphine auxiliary ligands [34]. They put forward the important argument that although using the sum of the van der Waals argument is the common criterion to assign argentophilicity, it is also critical to consider subtle electronic modulation that can be achieved by the auxiliary ligands which can ultimately play a key role in controlling the geometry and nuclearity of the complexes, and this has obvious consequences in describing the bonding situation.

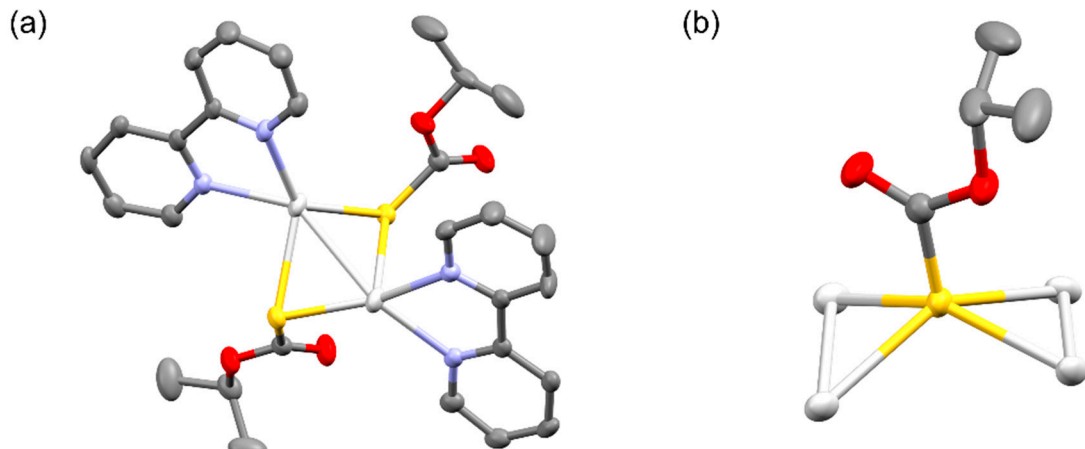

**Figure 3.** (**a**) Representation of **1** showing two monothiocarbonate ligands bridging two Ag(I) centres to form a Ag(I)···Ag(I) unit. (**b**) Representation of **1** showing the tetrametallic tetraconnective coordination mode of the ligand bridging two Ag(I)···Ag(I) units. Colour code: white = silver, yellow = sulphur, red = oxygen.

The S–(Ag(I)···Ag(I))–S unit (Figure 3a) was similar to what has been observed in silver thiolate clusters and polymers [35–37]. In mixed dithiocarbamate/PPh$_3$ complexes, for example, dinuclear complexes with the same four-membered ring form [38] but these complexes are discrete dinuclear units and the dithiocarbamate ligands adopt a different coordination mode ($\kappa_3$: $\mu_2$-S, $\mu$-S'), resulting in shorter Ag(I)···Ag(I) distances.

The C–S, C=O, and C–O bond lengths in **1** were 1.755 Å, 1.205 Å, and 1.333 Å, respectively. These values indicate that double bond character was mostly localised on the C=O group, with the negative charge residing on the S atom. This was confirmed by the FTIR spectrum, which showed bands at 1653 cm$^{-1}$ and 1642 cm$^{-1}$, corresponding to C=O stretching frequencies, and bands at 1145 cm$^{-1}$ and 1083 cm$^{-1}$ were due to C–O and C–S stretching vibrations.

The $^1$H NMR spectrum of **1** in CDCl$_3$ showed that the 2,2'-bpy ligand protons resonated at 8.64, 8.28, 7.78 and 7.27 ppm. The methyl protons resonated as a multiplet at 1.17 ppm, while the C-O-CH proton resonated as a triplet at 4.79 ppm. The ratios obtained by integration of the NMR peaks suggested that the ratio of 2,2'-bpy ligands to monothiocarbonate ligands is 1:1 in chloroform solution. This suggested that **1** exists in solution as discrete mono- or oligomeric units, tentatively formulated as [Ag(S(O)CO$^i$Pr)(2,2'-bpy)]$_n$ (n = 1,2, 3 . . . ), and we therefore propose that in solution, complex **1** experience a degree of dissociation, but the degree of oligomerization at equilibrium cannot be determined with accuracy based only on the NMR data.

In the solid-state, complex **1** is a coordination polymer with short Ag(I)···Ag(I) contacts. When placed under a UV lamp (365 nm) at room temperature, the solid showed intense yellow emission. The emission maximum was at 526 nm when the sample was excited at 368 nm (Figure 4). The intense luminescence observed for **1** can be attributed to metal-centred (d-s/d-p) transitions (due to the presence of Ag(I)···Ag(I) interactions) and LMCT transitions [39,40]. While LCCT transitions on [SC(O)O$^i$Pr] are not expected, LCCT transitions on 2,2'-bpy likely contribute to the luminescence of **1**. When the solid was dissolved in chloroform, however, emission became weak, presumably due to partial decomposition and/or dissociation as mentioned above.

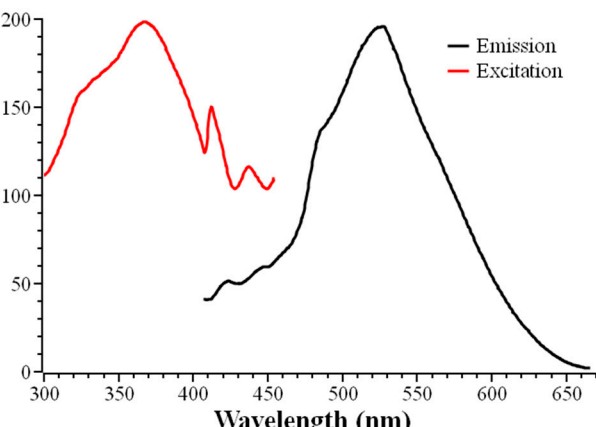

**Figure 4.** Solid-state luminescence spectrum of **1** with an emission maximum at 526 nm at an excitation of 368 nm.

### 2.2. The Gold(I) Complex [Au₂{S(O)CO^iPr}₂(dppe)]ₙ 2

In 2017, two gold(I) monothiocarbonate complexes [Au{S(O)COR}(PPh₃)₂] (R = ^iPr, ^iBu) were reported [21]. Both complexes were mononuclear with distorted trigonal planar Au(I) centres, and the monothiocarbonate ligands adopted a monodentate μ-S coordination mode. The complex [Au₂{S(O)CO^iPr}₂(dppe)]ₙ **2**, was synthesised in a two-step procedure wherein ClAu(THT) (THT = tetrahydrothiophene) was reacted with 0.5 molar equivalents of 1,2-bis(diphenylphosphino)ethane (dppe) to give Au₂Cl₂(dppe), which was isolated as a white solid before K[SC(O)O^iPr] was introduced. Complex **2** was isolated as a white powder, and single crystals were obtained by slow diffusion of diethyl ether into a concentrated dichloromethane solution at 4 °C. SCXRD analysis revealed that **2** crystallised in the monoclinic P2₁/n space group. The asymmetric unit consisted of two [Au₂{S(O)CO^iPr}₂(dppe)] units held together by a short Au(I)···Au(I) interaction (Figure 5). The relevant crystallographic data for complex **2** is summarised in Table 1.

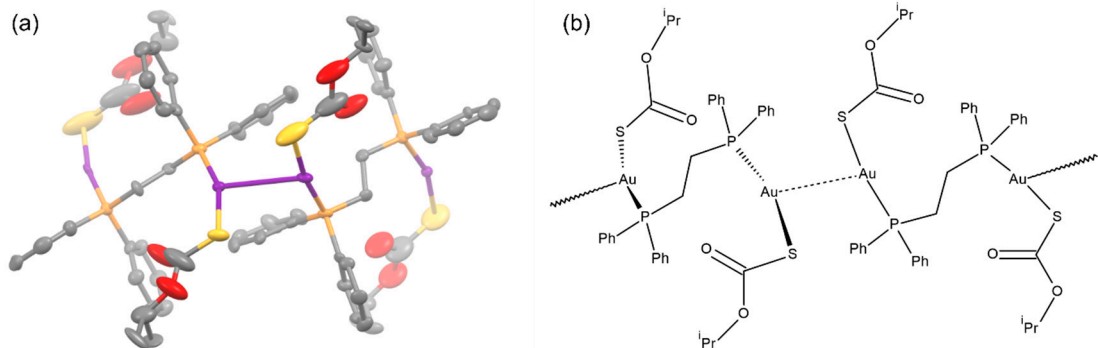

**Figure 5.** (**a**) Molecular structure of the asymmetric unit of **2**. All hydrogen atoms have been omitted for clarity. Thermal ellipsoids are shown at 50% probability. (**b**) Structure drawing of the asymmetric unit, which consisted of two [Au₂{S(O)CO^iPr}₂(dppe)] units joined by an aurophilic interaction.

Complex **2** exists as a 1-dimensional chain structure of the formula [Au₂{S(O)CO^iPr}₂(dppe)]ₙ. Aurophilic interactions allowed formation of [···Au(I)-P(Ph₂)CH₂CH₂P(Ph)₂-Au(I)···] chains with one monothiocarbonate ligand coordinated to each Au(I) centre to form branches off the chain. Each monothiocarbonate ligand coordinated via a monodentate μ-S bonding mode, as was seen in the structure of **1**.

The coordination geometry around the Au(I) centres was approximately linear, with the coordination sites occupied by one P atom and one S atom. This geometry left the Au coordination sphere open for close interactions with neighbouring Au(I) centres. A structural representation of the aurophilic interaction between two Au(I) centres is shown

in Figure 6. This interaction was unsupported, and the Au(I)···Au(I) distance was 3.21 Å, which is significantly shorter than the sum of two van der Waals radii (3.80 Å), indicating an aurophilic Au(I)···Au(I) interaction. The linear coordination geometry around each Au(I) centre was significantly distorted as the coordinating P and S atoms appear to be pressed together, distorting the P–Au–S bond angles to 160° (P1–Au1–S1) and 170° (P2–Au2–S2). The two P–Au–S units were oriented in a staggered orientation (the P1–Au1–Au2–P2 torsion angle was 96°), most likely to minimise unfavourable steric interactions.

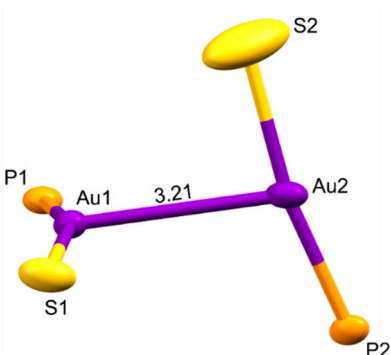

**Figure 6.** Structural representation of the unsupported aurophilic interaction in **2**. (Au(I)···Au(I) distance = 3.21 Å).

There are numerous reported gold(I) complexes with thiolate-type ligands, including those derived from dithiocarbamates and monoanionic thioureas [41–45]. The crystal structure of **2** was similar to that of [$Au_2$(dppe)(D-Hpen-S)$_2$], reported by Lee et al. [46], where D-Hpen = D-penicillaminate. In [$Au_2$(dppe)(D-Hpen-S)$_2$], the coordination geometry around the Au(I) centres was approximately linear with an average S–Au–P angle of 172.9°, and the average Au–S and Au–P bond lengths were 2.305 Å and 2.263 Å. In **2**, the Au–S and bonds Au–P were and 2.264–2.358 Å and 2.241–2.246 Å, respectively.

The distance between Au(I) centres and carbonyl O atoms of [SC(O)O$^i$Pr] ranged from 3.00 Å to 3.29 Å. The lower end of the range of distances suggests possible weak Au(I)···O interactions since these are slightly less than the sum of two van der Waals radii of the atoms involved [47]. Such interactions have been observed in several other gold complexes [48–51].

The dppe ligands in **2** adopted the *anti* conformation, and most of the [{Au(SR)}$_2${dppe}] complexes that have been reported to date have dppe ligands in this *anti* conformation [41,52]. A noteworthy exception is the thiocarbamide-containing [{Au(SR)$_2$}{dppe}] complex that was isolated by Ho et al., in which the dppe ligands adopted the *syn* conformation to maximise intramolecular Au(I)···Au(I) interactions [50].

Lee et al. noted that [$Au_2$(dppe)(D-Hpen-S)$_2$] was photoluminescent in the solid-state and reported strong emission at 524 nm when excited at 290 nm at room temperature [46]. The emission behaviour of **2** was studied both in solid-state (Figure 7.) and in dichloromethane solution (Figure 8). The emission spectrum of **2** showed a sharp peak at 480 nm when excited at 400 nm (this was the excitation maximum observed when emission at 480 nm was studied), and shoulder peaks at ~520 nm and 600 nm. The mononuclear complexes [Au{S(O)CO$^i$Pr}(PPh$_3$)$_2$] and [Au{S(O)CO$^i$Bu}(PPh$_3$)$_2$] showed emission maxima at 462 nm and 471 nm, respectively [21]. The emission peak at 480 nm was therefore tentatively attributed to metal-centred and LMCT transitions.

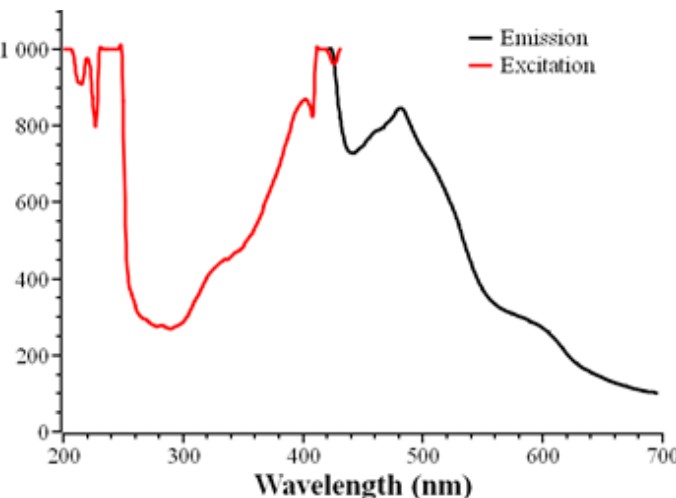

**Figure 7.** Solid-state photoluminescence spectrum of **2** with an emission maximum at 480 nm at an excitation of 400 nm. The peak at 230–250 nm in the excitation spectrum was an artefact which consistently appeared at $\lambda = \frac{1}{2}\lambda_{em}$.

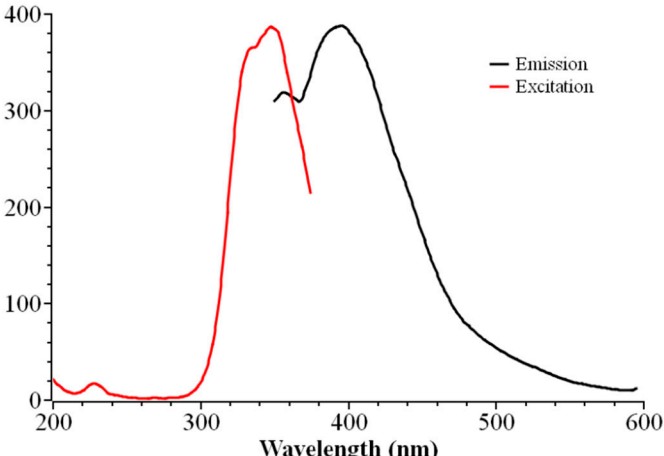

**Figure 8.** Photoluminescence spectrum of **2** in dichloromethane.

When complex **2** was dissolved in dichloromethane, the emission and excitation spectra showed better peak profiles, with clear excitation and emission maxima at 350 nm and 395 nm, respectively (Figure 8). Evidently, dissolution in dichloromethane resulted in a significant blue-shift in the emission and excitation maxima ($|\Delta\lambda_{em,max}| = 85$ nm). The emissive behaviour of **2** in solution is surprising, since [Au{S(O)CO$^i$Pr}(PPh$_3$)$_2$] and [Au{S(O)CO$^i$Bu}(PPh$_3$)$_2$] were not emissive in solution, and no solution-phase emission was reported for [Au$_2$(dppe)(D-Hpen-S)$_2$]. The observation is also different from complex **1**, which showed only weak emission in chlorinated solvents.

While a general correlation between Au(I)···Au(I) distance and emission energy is more difficult to track, the effect of aurophilic interactions on luminescence has been clearly established. The shift in luminescence maxima could be due to a change in the aurophilic interaction that is present in the solid-state. The energy of an aurophilic Au···Au interaction has been determined to be in the range of a typical hydrogen bond [53], therefore, the aurophilic interactions are expected to be disrupted (or weakened) in solution. Since the mononuclear complexes reported by Cyue et al. [21] did not exhibit luminescent behaviour in solution, the origin of the emission in dichloromethane solutions of **2** is likely due to weak aurophilic interactions that are still present in solution.

## 3. Materials and Methods

### 3.1. General

Synthesis of the complexes was performed with standard Schlenk techniques. Hexafluorophosphoric acid, KOH and $Ag_2O$ were purchased from Sigma Aldrich and used without further purification. Gold was received from Rand Refineries as a solution in aqua regia. Acetone, acetonitrile, chloroform, and diethyl ether were purchased from Honeywell. Hexane was purchased from ACE Chemicals. Propan-2-ol and was purchased from Merck. All solvents were used as received, without further drying or purification. $[Ag(CH_3CN)_4]PF_6$ was prepared using a method based on the literature procedure for the synthesis of $[Cu(CH_3CN)_4]PF_6$ [54]. ClAu(THT) was prepared using literature methods [55].

NMR spectra were recorded at 298 K on a Bruker Avance 400 MHz spectrometer. Residual proton impurity in the deuterated solvents was used for referencing of [1]H and [13]C spectra. FTIR spectra were recorded on a Perkin-Elmer Spectrum 100 FTIR spectrometer equipped with a Universal ATR Accessory. Photoluminescence spectra were recorded at ambient temperature on a Perkin-Elmer LS-55 spectrometer fitted with a front surface accessory.

### 3.2. Synthesis of K[S(O)COiPr]

A mixture of 2-propanol (35 mL, excess) and powdered KOH (18.35 g, 0.328 mol) was stirred at 40 °C for 1 h. After cooling the solution to ambient temperature, 5 mL deionised $H_2O$ was added to dissolve remaining solids. COS gas was generated in a separate flask using a method adapted from that described by Demselben [56]. Briefly, a stock solution of 55% (*w/w*) $H_2SO_4$(aq) was prepared using 98% $H_2SO_4$ and deionised water. To generate an excess of COS(g), 142.66 g (0.4 mol) of this stock solution was added to KSCN (38.87 g, 0.4 mol) and the mixture heated to 40 °C. The COS(g) formed was directed to the 2-propanol/$H_2O$ solution of potassium isopropoxide using a cannula. After stirring at ambient temperature for 2 h, 200 mL of n-hexane was added to the reaction flask and the mixture was stirred in an ice bath for 1 h. K[S(O)COiPr] was isolated as a white solid by vacuum filtration and washed with 25 mL hexane and 10 mL diethyl ether before drying *in vacuo* (10.69 g, 0.0688 mol, 21% yield—based on KOH). [1]H NMR (400 MHz, $D_2O$) δ (ppm) 1.14 (d, $CH_3$, 6H), 4.77 (q, O-CH-C, 1H). [13]C NMR (400 MHz, $D_2O$) δ (ppm) 185.02, 70.33, 21.18. FTIR: ($cm^{-1}$) 1585 (C=O), 1127 (C–O), 1073 (C–S).

### 3.3. Synthesis of Complex [{Ag$_4$(SC(O)O$^i$Pr)$_2$(2,2′-bpy)$_4$}(PF$_6$)$_2$]$_n$ 1

Precursor $[Ag(CH_3CN)_4]PF_6$ (658.8 mg, 1.580 mmol) was transferred to a dry Schlenk tube and suspended in 10 mL acetone. K[S(O)CO$^i$Pr] (250.0 mg, 1.580 mmol) and 2,2′-bpy (246.7 mg, 1.580 mmol) were dissolved in 15 mL acetone. The solution of the ligands was added to the Schlenk tube in one portion and the reaction mixture stirred 1 h. The mixture was filtered and the solvent was removed from the filtrate in vacuo. The light brown solid residue was extracted with DCM and filtered over celite. Colourless crystals of **1** were obtained by concentration of the DCM extract and cooling to 4 °C. [1]H NMR (400 MHz, $CDCl_3$) δ(ppm) 1.17 (m, $CH_3$, 6H), 4.79 (t, O-CH-CH, 1H), 7.27 (m, Ar-H, 2H), 7.78 (m, Ar-H, 2H), 8.28 (dd, Ar-H, 2H), 8.64 (t, Ar-H, 2H). FTIR: ($cm^{-1}$) 1654, 1592 (C=O), 1147 (C–O), 1084 (C–S). Elemental analysis calculated for $C_{48}H_{46}Ag_4N_8O_4S_2F_{12}P_2$: C 36.39%, H 2.93%, N 7.07%, found: C 36.16%, H 2.88%, N 6.96%.

### 3.4. Synthesis of [Au$_2${S(O)CO$^i$Pr}$_2$(dppe)]$_n$ 2

A solution of 1,2-bis(diphenylphosphino)ethane (155.4 mg, 0.3899 mmol) in dry DCM was added to ClAu(THT) (251.0 mg, 0.7798 mmol) in an oven-dried Schlenk tube. The mixture was stirred for 30 min. The solvent was removed, and the remaining white solid was left under vacuum overnight. The product, $Au_2Cl_2$dppe, was isolated but not further purified or characterised. Acetone (15 mL) was added to the Schlenk tube containing $Au_2Cl_2$dppe (0.3899 mmol, quantitative yield was assumed). The mixture was stirred to

form a cloudy mixture before K[S(O)CO$^i$Pr] (123.1 mg, 0.7778 mmol) was added as a solid. A few drops of 2-propanol were added to improve the solubility. The mixture was stirred at room temperature for 3 h before a white/grey precipitate was removed by filtration. The solvent was removed from the filtrate, and the residue was extracted using DCM. After filtration through celite, the DCM filtrate was concentrated, and this solution was used directly for crystallisation. Single crystals of **2** were obtained from slow diffusion of diethyl ether into the concentrated DCM solution at 4 °C. $^1$H NMR (400 MHz, CDCl$_3$) δ (ppm) 1.13–1.21 (two doublets, O-CH$_3$, 12H), 2.64 (s, P-C$_2$H$_4$-P, 4H), 3.94 (m, O-CH-C, 1H), 4.93 (m, O-CH-C, 1H), 7.43 (m, Ph-H, 12 H), 7.64 (q, Ph-H, 8H). $^{13}$C NMR (400 MHz, CDCl$_3$) δ (ppm) 22.08 (P-C$_2$H$_4$-P), 25.37 (O-CH$_3$), 70.34 (O-CH-C), 129.64 (Ph), 132.46 (Ph), 133.41 (Ph). FTIR: (cm$^{-1}$) 1638 (C=O), 1135 (C–O), 1094 (C–S). Elemental analysis calculated for C$_{34}$H$_{38}$Au$_2$O$_4$P$_2$S$_2$: C 39.62%, H 3.72%, found: C 39.98%, H 3.79%.

*3.5. Crystallography*

Single crystals were mounted on a glass fiber with a small amount of Paratone$^®$ oil (Sigma Aldrich, MO, USA). Intensity data was collected at 150 K on a Bruker SMART APEX II diffractometer with an APEX II CCD area detector. The instrument was equipped with graphite monochromated Mo-K$\alpha$ X-ray source and an Oxford Cryosystems Cryosystem Controller 700 (Oxford Cryosystems Ltd., Long Hanborough, UK). Data reduction and absorption correction were carried out using SAINT-Plus [57] and SADABS [58] software, respectively. The structures were solved by direct methods using SHELXS [59] and refined by full-matrix least-squares on $F^2$ using the SHELXL [60] software package in OLEX$^2$ [61] Illustrations of crystal structures were generated in Mercury [62] using POV-Ray (in Supplementary Materials).

**4. Conclusions**

In summary, the synthesis of a silver(I) and gold(I) monothiocarbonate complex are described. This is the first silver(I) complex of this nature reported, and the gold(I) complex is new. The complexes were characterized by single crystal X-ray crystallography and NMR. The coordination environment of the Ag(I) centres was unique forming a coordination polymer, while the Au(I) complex revealed an infinite one-dimensional chain structure that formed by the association of [Au$_2${S(O)CO$^i$Pr}$_2$(dppe)]$_n$ molecules through aurophilic interactions. Both complexes were emissive in the solid state, whilst for silver the emission was weakened in chlorinated solvent, for gold(I) the emission remained strong. In both complexes, respective metallophilic interactions were present. These ligands have significant potential in further exploration of the Group 11 triad in terms of structural flexibility, reactivity, and solid-state luminescence.

**Supplementary Materials:** The following are available online at https://www.mdpi.com/article/10.3390/inorganics10020019/s1, FT-IR spectra for the ligand, and complexes **1** and **2**. NMR data for the ligand and compexes **1** and **2**. Crystallographic data for **1** and **2**, (7 Tables each), CIF and CheckCIF files of **1** and **2**.

**Author Contributions:** Conceptualization, W.D., M.N.P. and W.E.v.Z.; methodology, W.D. and M.N.P.; validation, W.D.; formal analysis, W.D., M.N.P. and W.E.v.Z.; investigation, W.D. and M.N.P.; writing—original draft preparation, W.D. (as part of MSc thesis); writing—review and editing, W.E.v.Z.; visualization, W.D.; supervision, W.E.v.Z.; project administration, W.E.v.Z.; funding acquisition, W.E.v.Z. All authors have read and agreed to the published version of the manuscript.

**Funding:** W.E.v.Z. is grateful to the University of KwaZulu-Natal and the Eskom TESP Programme for funding part of this work.

**Institutional Review Board Statement:** Not applicable.

**Informed Consent Statement:** Not applicable.

**Data Availability Statement:** FT-IR and NMR spectra and crystal data are given in the supporting information. Crystal data, details of the data collection and refinement are given in for complexes **1**

and **2** (7 Tables each). Crystallographic data for the structures have been deposited with the Cambridge Crystallographic Data Centre (CCDC no. 2126277 and 2126328). Copies of this information may be obtained free of charge from The Director, CCDC,12 Union Road, Cambridge, CB2 1EZ, UK (fax: +44-1223-336033; email: deposit@ccdc.cam.ac.uk or http://www.ccdc.cam.ac.uk accessed on 3 January 2022).

**Conflicts of Interest:** The authors declare no conflict of interest. The funders had no role in the design of the study; in the collection, analyses, or interpretation of data; in the writing of the manuscript, or in the decision to publish the results.

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
