# Peer review of "Silver(I) and Gold(I) Monothiocarbonate Complexes: Synthesis, Structure, Luminescence"

_inorganics, doi:10.3390/inorganics10020019_

Round 1
Reviewer 1 Report
The manuscript entitled "Silver(I) and Gold(I) Monothiocarbonate Complexes: Synthesis, Structure, Luminescence " by Welni Duminy et. al. is anice piece of work and contribute significantly in the area. The work presented is of general interest of readers and I recommend the article for favor of publication in "Inorganics" before the i would like to suggest the author to take care of following points in revised manuscript-
- Authors are advised to have a look on grammatical errors in whole manuscript. like P9 last line “ biut”I think it would be "But"
- In order to enrich the content of article authors are advised to add Thermogravimetric analysis like TG/DTA in detail.
- PXRD of as synthesized complex which may ensure the phase purity of product, is missing please add.
- In the manuscript, solid state photoluminescence study of ligand is missing and compare it with complex in detail.
Author Response
We thank the reviewer for the diligent reading of our work and for suggestions to improve the paper.
Authors are advised to have a look on grammatical errors in whole manuscript. like P9 last line “ biut”I think it would be "But"
Response: This has been done and the problem(s) corrected
In order to enrich the content of article authors are advised to add Thermogravimetric analysis like TG/DTA in detail.
Response: The paper’s main focus is on the molecular and crystal structure of new complexes with possible structure-(optical)property relations pointed out. TG/DTA looks more at the material side which we did not pursue here.
PXRD of as synthesized complex which may ensure the phase purity of product, is missing please add.
Response: We performed satisfactory CHN elemental analysis to ensure bulk purity of the as-prepared samples and together with solution NMR, we are confident and satisfied with the phase purity.
In the manuscript, solid state photoluminescence study of ligand is missing and compare it with complex in detail.
Response: In this instance, the luminescence arises from the assembly/coordination where two different ligands are involved in each case. Thus, a report on the luminescence of each ligand would not be comparable, or provide a reasonable basis for discussion. Additionally, the free monothiocarbonate ligand is not emissive at all (no luminescence) in the solid state. Importantly, the lack of an emissive state in solution for the gold(I) coordination polymers, further emphasizes the solid-state assembly and combination of the ligands are crucial for stabilizing an emissive state.
Reviewer 2 Report
The manuscript by Werner E. van Zyl et al. describes two monothiocarbonate-based Ag(I) and Au(I) complexes bearing 2,2’-bipy or dppe co-ligands. These complexes have been synthesized and systematically studied exploiting various experimental methods. Moreover, the luminescence of the designed compounds was also studied. The reported findings contribute to the coordination chemistry of monothiocarbonate ligands and Group 11 metal complexes. In my opinion, the reviewed work is high quality and of the proper impact/scope for this journal. Thus, I recommend this paper be published in this journal after minor revision to address the following concerns.
Figure 4 displays both excitation and emission spectra. Meanwhile, the legend is named as “Solid-state photoluminescence spectrum of 1.” only. Please, correct this issue.
2. In the experimental part, the authors wrote that “powdered KOH” was used. It was actually anhydrous KOH? Commonly, brand potassium alkali contains about 15% water, and thereby formulated as KOH*0.5H2O. Please check this issue.
3. Figures 4 and 7: Please add to the legends the excitation energy that was used for recording emission spectra.
4. What about the emission quantum yield of the compound obtained? This information will be useful for readers dealing with luminescent materials.
5. If possible, please kindly add in the citation list the following relevant works on luminescent Ag(I) complexes containing S-donor ligands and short Ag…Ag contacts, in particular: 10.1039/C7DT00740J, 10.1016/j.inoche.2019.107513, 10.1007/s11243-021-00457-5, 10.1002/chem.201404049, 10.1016/j.ica.2019.01.036, 10.1039/D1NR02540F, 10.1002/anie.202100965, 10.1039/c9qi01069f.
Reviewer 3 Report
The manuscript “Silver(I) and Gold(I) Monothiocarbonate Complexes: Synthesis, Structure, Luminescence” by Van Zyl and collaborators presents the synthesis and characterization of two new silver(I) and gold(I) derivatives bearing isopropyl monothiocarbonate. The crystal structures were determined showing that, in contrast with the related dithiocarbonates or xanthates, this ligand acts preferentially as a monodentate thiolate, and the oxygen is barely coordinating to the soft Au(I) and Ag(I) centers. The structures are analyzed in the context of some literature-related compounds. Finally, the Luminescent behavior of the compounds is presented. The study seems correctly performed although it can be improved prior to publication in inorganics, following are my suggestions:
The sources for the considered van der Waals radii should be specified.
In the discussion of the structure of the polymeric structure 1, the existence of argentophilic interactions is proposed on the basis of interatomic distance considerations, the incidence of such interactions should be done in a more careful way, particularly taking into consideration the geometrical consideration recently raised by Caballero et al. Eur. J. Inorg. Chem. 2021, 2021, 2702 https://doi.org/10.1002/ejic.202100336.
Concerning the solution identity of 1, the authors propose a monomeric species based on the NMR integration. However, any oligomer [Ag(S(O)COiPr)(2,2’-bpy)]n regardless of the value of n would give the same integration ratio.
Regarding the gold(I) compound, it is also clear that this new type of ligand has a predominant thiolate ligand. The authors discuss the structure in the context of the available literature, but they let aside a number of documents in which similar structures are observed:
For example, while in the statement “The crystal structure of 2 was similar to that of [Au2(dppe)(D-Hpen-S)2], reported by Lee et al.,[30] where D-Hpen = D-penicillaminate.” one reference is mentioned there are several other similar structures reported, for instance:
- Organomet. Chem. 2005, 690, 57–68 https://doi.org/10.1016/j.jorganchem.2004.08.028 (cited in ref 35)
- RSC Adv., 2015, 5, 34992-34998 https://doi.org/10.1039/C5RA01831E
- Dalton Trans., 2011, 40, 589-596 https://doi.org/10.1039/C0DT00620C
- New J. Chem., 2018, 42, 7845-7852 https://doi.org/10.1039/C7NJ04354F
- Organomet. Chem. 2006, 359, 204–214 https://doi.org/10.1016/j.ica.2005.07.046
The photophysical measurements should be better explained. In particular, whether the solid-state measurement is referred to single-crystal or powder samples. If they were powder, a proof of phase purity should be included to show that the crystalline structures presented are representative of the bulk powder (PXRD). Furthermore, the emission used for acquiring the excitation spectra should be informed. In the case of solution measurements, the concentration must be clearly stated. Furthermore, please inform of any other correction applied to the spectra and the use of filters in the experimental section.
Finally, please proofread searching typos, in page 9 last line says biut
Round 2
Reviewer 1 Report
Authors have revised the manuscript but didn't consider my comments life
- Kindly include luminescence study of ligand too.
- CHN is good study but not as appropriate as PXRD so ensure the phase purity PXRD is, required one.
I can recommend thr article fir publication only after taking into consideration of above mentioned points in revised manuscript.
Author Response
We thank the reviewer for the comments.
- The question on the luminescence has been asked and explained in the previous response. The answer is the title free ligand used i.e. K[S(O)COR] shows NO luminescence under any excitation, and there is thus nothing to report further with regards to luminescence of the free ligand. The bpy (for Ag(I) and dppe (for Au(I)) ligands are well established in the literature and are not the cause for the emission of the present study, which we ascribe to the closed-shell d10 system for both metal ions. A thiolate system generally shows an emission that arises from a S-Au (ligand to metal) charge-transfer transition, LMCT, and when substituents on the sulfur ligands do not produce significant electronic perturbations, emission bands become strongly influenced by the intermetallic metal-metal interactions and distances.
2. We agree that the more samples can be analyzed by different techniques the better. Bulk powder purity is widely acknowledged to be acceptable and affirmed by either CHN analysis which we did, or by high-resolution mass spectrometry, and to a far less extent by PXRD which is typically done for comparison purposes with a literature reference. In that regard it is not clear how PXRD is going to establish anything not already known, either in bulk or molecular level.